# Small Subcutaneous Soft Tissue Tumors (<5 cm) Can Be Sarcomas and Contrast-Enhanced Ultrasound (CEUS) Is Useful to Identify Potentially Malignant Masses

**DOI:** 10.3390/ijerph17238868

**Published:** 2020-11-28

**Authors:** Armanda De Marchi, Simona Pozza, Lorena Charrier, Filadelfo Cannone, Franco Cavallo, Alessandra Linari, Raimondo Piana, Irene Geniò, Paolo Balocco, Alessandro Massè

**Affiliations:** 1Department of Imaging, Azienda Ospedaliero Universitaria Città della Salute e della Scienza, CTO Hospital, Via Zuretti 29, 10126 Torino, Italy; armanda.demarchi@tiscali.it (A.D.M.); simona.pozza@tin.it (S.P.); pbalocco@cittadellasalute.to.it (P.B.); 2Department of Public Health and Pediatrics, University of Turin, Via Santena 5-bis, 10126 Torino, Italy; franco.cavallo@unito.it; 3Radiology Department, Azienda Sanitaria Provinciale di Siracusa, E. Muscatello Hospital, Contrada Granatello, 96011 Augusta, Italy; delfo.cannone@gmail.com; 4Department of Pathology, Azienda Ospedaliero Universitaria Città della Salute e della Scienza, CTO Hospital, Via Zuretti 29, 10126 Torino, Italy; alinari@cittadellasalute.to.it; 5Department of Orthopaedic, Traumatology and Rehabilitation, Azienda Ospedaliero Universitaria Città della Salute e della Scienza, CTO Hospital, Via Zuretti 29, 10126 Torino, Italy; rpiana@cittadellasalute.to.it (R.P.); alessandro.masse@unito.it (A.M.); 6Department of Imaging, Azienda Ospedaliero Universitaria G. Martino, Via Consolare Valeria 1, 98100 Messina, Italy; genio.irene@gmail.com

**Keywords:** contrast media, ultrasonography, magnetic resonance imaging, soft tissue neoplasms, standards

## Abstract

Subcutaneous masses smaller than 5 cm can be malignant, in contrast with the international guidelines. Ultrasound (US) and magnetic resonance imaging (MRI) are useful to distinguish a potentially malignant mass from the numerous benign soft tissue (ST) lesions. Contrast-enhanced ultrasound (CEUS) was applied in ST tumors, without distinguishing the subcutaneous from the deep lesions. We evaluated CEUS and MRI accuracy in comparison to histology in differentiating malignant from nonmalignant superficial ST masses, 50% smaller than 5 cm. Sensitivity, specificity, and positive and negative predictive values (PPV, NPV) with their 95% confidence intervals (CI) were calculated. Of malignant cases, 44.4% measured ≤5 cm. At univariate analysis, no statistically significant differences emerged between benign and malignant tumors in relation with clinical characteristics, except for relationship with the deep fascia (*p* = 0.048). MRI accuracy: sensitivity 52.8% (CI 37.0, 68.0), specificity 74.1% (CI 55.3, 86.8), PPV 73.1% (CI 53.9, 86.3), and NPV 54.1% (CI 38.4, 69.0). CEUS accuracy: sensitivity 75% (CI 58.9, 86.3), specificity 37% (CI 21.5, 55.8), PPV 61.4% (CI 46.6, 74.3), and NPV 52.6% (CI 31.7, 72.7). CEUS showed a sensitivity higher than MRI, whereas PPV and NPV were comparable. Also, masses measuring less than 5 cm can be malignant and referral criteria for centralization could be revised.

## 1. Introduction

The soft tissue (ST) masses located in the subcutaneous tissue, between the skin and the superficial muscular fascia, are common in clinical practice. The histological diagnosis is wide, ranging from inflammatory and tumor-like lesions to ST sarcomas, lymphomas, skin appendage lesions, and metastatic tumors [1,2].

The superficial ST sarcomas are rare in comparison to numerous benign lesions but have peculiar clinical problems. First, the prognosis: in large series of cases [3,4,5], the 5-year and 10-year overall survival and metastasis-free survival were better than for deep tumors, but the local recurrence remained high, suggesting that the size and an initial adequate wide surgical *en bloc* excision are fundamental [3,4,5,6]. Second, the anatomical compartment: Sarcomas developed in the subcutaneous tissue, defined as “intracompartmental”, in absence of anatomical barriers, can spread in the subcutaneous compartment in all directions [1], often with an infiltrative pattern [3,4]. Third, the size: A significant proportion of malignant superficial sarcomas measure less than 5 cm in maximal diameter [3,4,5,6,7], whereas the current international guidelines identify a size larger than 5 cm and a site deep with respect to the muscular fascia as the main risk factors for malignancy [8,9]. 

These critical factors explain why the main challenge in everyday clinical practice is to use a sensitive and specific imaging technique to distinguish a potentially malignant mass from the numerous benign lesions located in the subcutaneous tissue. 

Considering age (adults vs. children and adolescents), anatomic site (epithelioid sarcomas usually occur in the hand and wrist), and the specific location within the superficial tissue, a diagnosis may be advanced with the help of imaging [2]. 

Grayscale ultrasonography (US) and power Doppler US (PD) are the main first-level imaging techniques for assessing superficial soft tissue masses. Their sensitivity and specificity can range from 70% to 100%. While some lesions, such as lipomas, vascular malformations and epidermoid cysts, have well-recognized ultrasound features, others can be misdiagnosed [2,10,11,12,13,14]. 

Magnetic resonance imaging (MRI) identifies the exact extension and the involved compartment and defines the tissue characteristics [4,6,13,15,16,17,18,19,20,21,22]. Without distinguishing the superficial from the deep ST tumors, literature reports MRI’s sensitivity and specificity in distinguishing benign from malignant tumors ranging, respectively, from 80% to 94% and from 64% to 94% [15,16,23]. In these papers, though, reported cases were mainly easily diagnosable tumors, like lipoma, cyst, and angioma [23]. Some MRI’s characteristics seem more prone to suggest malignancy. For example, Chen et al. [15] demonstrated, in 118 lesions, that the risk factors for malignancy at MRI were the combination of necrosis, maximal mass diameter, peritumoral edema, absent fibrosis, calcification, and fat rim. Considering the studies focused only on superficial tumors, other characteristics were emphasized. Galant et al. [24], in 64 subcutaneous soft tissue masses, demonstrated that malignant tumors have a higher tendency to develop a close relationship with the fascia than benign lesions and that an obtuse angle between the fascia and the mass, or a lesion crossing the fascia, strongly suggests malignancy. Calleja et al. [6] studied, retrospectively, 136 patients with US and 72 also with MRI. They found a significant relationship between malignancy and lobulation, hemorrhage, fascial edema, necrosis, skin thickening, and contact, but not with size. 

Contrast-enhanced US (CEUS) consists of an injection of gas-filled microbubbles, able to depict in real time the microvascularity and neoangiogenesis of the lesion, increasing the confidence in diagnosing a tumor. Three studies reported some preliminary results concerning the ST tumors studied by CEUS [25,26,27]. Without distinguishing between superficial and deep masses, two studies (respectively, on 54 and 216 ST lesions) demonstrated that inhomogeneous perfusion, due to central tumoral necrosis, and arterial uptake were associated with high risk of malignancy [26,27], whereas size and the relationship with the muscular fascia were predictive factors just in one study [26] and not in the other [27]. The third study [25] on 25 superficial masses demonstrated that three-dimensional power Doppler ultrasound with echo-contrast medium is a valuable tool for differential diagnosis of soft-tissue tumors.

We evaluated the accuracy of CEUS and MRI in differentiating malignant from nonmalignant tumors in 63 consecutive ST masses located in the subcutaneous tissue, referred at a regional sarcoma center, using as ‘gold standard’ the final histological diagnosis. 

## 2. Materials and Methods

### 2.1. Patients

The study group consisted of 63 consecutive cases with a subcutaneous lesion, defined as a mass located between the epidermidis and the muscular fascia. The patients were referred at a center for bone and soft tissue sarcoma from June 2012 to December 2014. Inclusion criteria: patients at least 18 years old, a marginated mass in the subcutaneous tissue studied with ultrasound (US), power Doppler (PD), contrast-enhanced US (CEUS), and magnetic resonance imaging (MRI), with histological diagnosis on bioptic sample and/or excisional sample. Exclusion criteria: children and adolescents, poorly marginated soft tissue masses (such as lipohypertrophy, soft tissue edema, or focal inflammation), purely epithelial mass, mass crossing the deep fascia, absence of CEUS imaging, and/or absence of histology.

All patients gave their informed consent to perform the imaging and to utilize their clinical data for study purposes. The local institutional board approved (protocol n. 0134022/2013) the study, which was carried out in compliance with the Code of Ethics of the World Medical Association.

### 2.2. Imaging

#### 2.2.1. US and Power Doppler Sonography (PD)

The US devices were Esaote My Lab Twice (Esaote, Genova, Italy 2011). The following characteristics were recorded: shape of the lesion (fusiform, round, serpiginous, lobulated, soap bubbles or cauliform, undefined), margins (regular well-defined or irregular infiltrating ill-defined), echostructure (isoechoic, anechoic, hyperechoic with anechoic areas, heterogeneous, or hypoechoic), calcifications (absent, present), septation (absent, present), location of the mass in the protective adipofascial system (PAFS) or in the lubricant adipofascial system (LAFS) or in both systems (according to Nakajima et al. [28], where the two adipofascial layers are delimitated by a thin superficial fascia well documented by US), relationship of the mass with the muscular fascia according to Galant et al. [24] (no contact, slight contact with acute angle between the tumor and the fascia, wider contact with larger acute or right angle, wide contact with obtuse angle with fascia; for statistical analysis, the first two and the second two variables were gathered together), and tissue characterization (fat lesion, cyst, vascular lesion). At power Doppler (PD), vascularization was recorded as absent or present.

#### 2.2.2. CEUS

CEUS scanning was performed with a low MI (mechanical index <= 0.1) technique and a dedicated software (CnTI Contrast Tuned Imaging. Esaote, Genoa, Italy). Scanning time of the lesions was documented in a continuous 3 min and on movie loop. Each patient received a dose of 4.8 mL of ultrasound contrast agent consisting of microbubbles filled with sulphur hexafluoride (SonoVue™, Bracco, Milan, Italy) via a 20-gauge intravenous cannula, followed by a flush of 5 mL saline solution. Five patients were treated with oral metylprednisolone and hydroxyzine dichlorhydrate in order to prevent any allergy in presence of a positive history; no allergic reactions occurred. CEUS pattern (P) was assessed according to De Marchi et al. classification [27], based on the morphology of the lesion and vessels’ distribution: P1, absence of contrast uptake; P2, enhancement only in the peripheral area of the lesion; P3, thin (<2 mm) and few vessels (<5/field); P4, thinner (>2 mm) and more numerous vessels (>5/field); P5, enhancement with a reticular aspect and both thick and thin bands inside; P6, numerous vessels, important and inhomogeneous enhancement with avascular areas; P7, numerous vessels in all areas, with homogeneous distribution. The vascularization time was recorded as >20 s (venous phase) and <20 s (arterial phase). During CEUS, qualitative perfusion analyses were performed at the maximum tumor enhancement and each mass was assigned to one of the seven defined perfusion patterns (P1–P7). The heterogeneous enhancement, P6, with avascular areas, is a typical feature of sarcoma [26].

#### 2.2.3. MRI

MRI was performed with General Electric Signa Exite 1,5 T. The studies included at least one T1-weighted spin echo and Short-TI Inversion Recovery (STIR) or fat-T2-suppressed weighted fast spin-echo sequences in a combination of axial, coronal, and sagittal plane, without and with the use of an intravenous paramagnetic contrast agent. The following features were assessed: signal intensity on T1- and T2-weighted images (isointense to subcutaneous tissue, homogeneous low signal intensity or hypointense, hyperintense with septa or high signal intensity, inhomogeneous low signal intensity or hypointense) and fat-saturated T2-weighted sequence or fat-suppressed sequence with inversion recovery pulse (STIR) (mass with homogeneous low signal intensity or hypointense, mass with inhomogeneous low signal intensity, mass with homogeneous high signal intensity; mass with inhomogeneous high signal intensity). After contrast medium injection, the data were recorded as absence of enhancement, homogeneous, or inhomogeneous enhancement. The peritumoral edema, the septation, and fat rim were recorded as absent or present. The margins were evaluated as regular and well-defined or irregular, infiltrating, and ill-defined. The tissue characteristics were recorded as fibrous tissue signal, fatty tissue, cystic signal, necrotic signal, myxoid tissue, vascular signal, void signal, or hemorrhage.

For both CEUS and MRI the hypothesis of benign or malignant tumor or mass of uncertain origin was assessed, in accordance with the literature [2,4,11,13,19,20,22,29,30], by three radiologists with at least 10 years of relevant experience in musculoskeletal radiological imaging.

### 2.3. Histology

Routine histopathological examination, immunohistochemistry, and molecular biology analyses were performed according to international guidelines. The 2012 World Health Organization (WHO) classification was adopted. All specimens were reviewed by two pathologists, expert in sarcoma and active partners of the interdisciplinary group.

### 2.4. Statistical Analysis

Descriptive data are shown as absolute frequencies of the different modalities for categorical data and as mean ± standard deviation (SD) for continuous variables.

The t-Student test for continuous variables and chi-square test for categorical variables were carried out to evaluate the association between patients’ and tumors’ characteristics at imaging with respect to histological assessment.

Accuracy of CEUS and MRI in comparison to histology as gold standard has been expressed as sensitivity, specificity, and positive and negative predictive values (PPV, NPV), together with their 95% Confidence Intervals (CI 95%). For MRI accuracy assessment, cases classified as uncertain were considered benign lesions, according to the literature [11].

A *p*-value of less than 0.05 was taken as significant. All analyses were performed with Stata 14 (StataCorp LLC, College Station, TX, USA).

## 3. Results

The mean age of our sample was 57.9 years (range 24–87 years). Gender was represented by 30 females (47.6%) and 33 males (52.4%). The nontumoral lesions were 10 (15.9%), the benign and locally aggressive intermediated tumors were 17 (27%), the malignant tumors (soft tissue sarcoma (STS), non-STS malignant tumors) were 36 (57.1%). The detailed histological diagnoses are reported in Table 1.

The masses were located at upper limbs in 15 cases (hand and wrist in three cases), at lower limbs in 37 cases (foot and ankle in four cases), at trunk in eight cases, and at pelvis in three cases. For statistical analysis, two main groups were considered: location in the upper part (trunk and upper limbs) and lower part (lower limbs and pelvis) of the body, respectively, 36.5% and 63.5% of all cases. The tumors were primary in 54 cases (85.7%) and recurrent in nine cases (14.3%).

Table 2 reports, in the first part, the descriptive data of the cases: gender, age, location, and primary vs. recurrent lesions. In the second part, US and PD data and the univariate analysis of benign vs. malignant tumors in relation with the histological diagnosis are reported.

Mean age was significantly higher among patients with malignant tumors (*p* = 0.006).

The size was ≤3.5 cm in 23.8%, between 3.6 and 5 cm in 27%, between 5.1 and 10 cm in 36.5%, and >10.1 cm in 12.7% cases. Respectively, 29.6% and 59.3% of benign and 19.4% and 44.4% of malignant tumors measured ≤3.5 cm and ≤5 cm. 

The main part of the lesions, independently from the diagnosis, had a fusiform shape (66.7%), regular margins (69.8%), septa (81%), and no calcifications (87.3%). 

Concerning the relationship with the fascia, the main part of the masses was located in both adipofascial systems (54%) and were in contact with the deep fascia with a large acute angle or an obtuse angle (55.5%). 

Concerning the echostructure, independently from the diagnosis, 76.2% of cases were heterogeneous or hypoechoic.

Considering the vascularization detected by PD sonography, 60.3% of total cases had a vascular supply.

At univariate analysis, no statistically significant differences emerged between benign and malignant tumors in any of the above-listed US and PD characteristics, except for the relationship with the deep fascia (*p* = 0.048).

Table 3 reports, in detail, CEUS and MRI data and the results at univariate analysis of benign vs. malignant tumors in relation with the histological diagnosis.

Gathering together patterns 6 and 7 as potentially malignant (69.8% of the cases) versus patterns 1 to 5 as potentially benign lesion (30.2% of the cases) did not yield statistically significant differences between benign and malignant tumors.

Vascularization time at CEUS was rapid (arterial phase), respectively, in 69.4% of malignant cases and 66.7% of benign tumors, without any statistically significant difference between the two groups.

All types of signal intensity on T1- and T2-weighted images and fat-suppressed sequence with inversion recovery pulse (STIR) were documented, without any statistically significant difference between benign and malignant tumors at univariate analysis.

Peritumoral edema was present in 31.7% of cases and a fat rim and regular margins in 69.8% of cases. Intravenous contrast agent was used in 41 patients: 87.8% of the cases showed gadolinium enhancement.

Septation was present only in four of 27 benign lesions (14.8%) and in 12 of 36 malignant tumors (33.3%) with no statistically significant difference.

At univariate analysis, no statistically significant difference emerged between benign and malignant tumors in any of the above-listed CEUS and MRI characteristics. As none of these characteristics has come out by itself as indicator of benignity or malignancy, the final diagnosis has been decided, as it happens in current practice, by expert radiologists, based on the evaluation of all the tumor’s characteristics emerging from the two imaging techniques.

Table 4 reports the accuracy in differentiating benign from malignant tumors in relation to histology based on MRI.

Table 5 reports the accuracy in differentiating benign from malignant tumors in relation to histology based on CEUS.

## 4. Discussion

Subcutaneous ST tumors are frequently observed in everyday clinical practice. US is the first-level imaging technique, accurate in identifying the more frequent benign lesions such as cysts, lipoma, bursitis, abscess, and hematoma [2,10,12,13,14,31,32,33]. Color and power Doppler can increase the US accuracy if the radiologist holds the probe with gentle contact with the skin in order to identify small vessels [10]. According to Hung and colleagues [12], the sensitivity and specificity of US for identifying malignant, superficial, soft tissue tumors were, respectively, 94.1% and 99.7%; however, they studied only 11 malignant tumors (only one sarcoma) out of 714 cases. Moreover, it is well known that US accuracy in differentiating malignant tumors from more frequent benign lesions decreases when the diagnosis of hematoma is involved [31]. MRI depicts the exact extension of the tumors, the tissue characteristics, and the relationship with the fascia [4,6,13,15,16,17,18,19,20,21,22]. The use of CEUS in soft tissue tumors has been used in few studies, mainly without differentiating superficial and deep lesions [25,26,27]. We evaluated the CEUS and MRI accuracy in comparison to histology in 63 purely subcutaneous tumors. 

In our sample, at univariate analysis, mean age was found significantly different in patients with malignant tumors (older people resulting in being more affected), whereas we were unable to demonstrate any significant difference in terms of gender and location between benign and malignant tumors, although other studies have shown some differences [4,6].

These results are partially in accordance with Chiou et al. [10], who did not find any significant difference between benign and malignant tumors in terms of echogenicity, composition, and Doppler features; however, they found significant differences in terms of margins, shape, and size.

Focalizing the attention on size, in our sample 44.4% of malignant cases measured <= 5 cm. At univariate analysis, the size did not result as a predictive risk factor for malignancy, confirming previous research [3,6,7]. Furthermore, recently, Kim and Chung demonstrated that the diagnostic accuracy of US-guided core needle biopsy in 500 small (mean size 5.1 cm) subcutaneous lesions (218 malignant) was higher for lesions <2 cm than for larger lesions [34]. However, the international soft tissue sarcoma guidelines identify only a mass larger than 5 cm and deep with respect to the muscular fascia as potentially malignant [8,9]. Our results confirm that also subcutaneous small tumors (<3.5 cm or <5 cm) can be potentially malignant [3,6,7] and that a revision of referral criteria for superficial soft tissue masses could be considered [6].

Considering MRI characteristics, in our sample none of the variables taken into account showed a statistically significant difference in discriminating benign from malignant tumors. Our results are partially in contrast with other studies in which, without distinguishing the superficial from the deep ST tumors, irregular infiltrative margins, septa and shape [10], T2 low signal matrix, calcification, necrosis, fat rim sign, peritumoral edema, and hemorrhage [15] are considered potential indicators of malignancy.

As far as CEUS is concerned, in our sample we did not find any statistically significant difference between benign and malignant tumors in term of CEUS pattern and time of vascularization. These results are in contrast with previous studies, which demonstrated that inhomogeneous enhancement with avascular areas and rapid vascularization time are related with malignancy [26,27]. However, these studies did not differentiate subcutaneous from deep lesions [26,27], whereas our sample was limited to the pure subcutaneous masses (Figure 1).

As for the accuracy in comparison with histology, MRI showed a low sensitivity (52.8%) in comparison to CEUS (75%): 17 cases out of 36 malignant tumors resulted in being negative at MRI while only nine at CEUS. This means that about 50% of malignant tumors turned out to be false-negative results at MRI, whereas only 25% were missed by CEUS. This is extremely important in considering this diagnostic process, as, in the case of very serious diseases like malignant tumors, what is most important from the very first steps of the diagnostic process is not to lose positive cases, while false-positives can be dealt with subsequently.

This low sensitivity of MRI in identifying malignant tumors is partially in contrast with the literature. Without distinguishing the superficial from the deep ST tumors, the reported sensitivity in the literature ranges from 75% to 94% and the specificity from 64% to 94% [15,16,17,23,30].

As for the positive predictive value, which is the endpoint of the diagnostic process in terms of efficiency of the procedure, MRI showed a good predictive value for a positive result (73.1%), higher than for CEUS (61.4%). However, the CI 95% showed no statistically significant difference between positive predictive values of MRI and CEUS.

CEUS specificity was obviously lower in comparison with MRI (37% vs. 74%), but the predictive value for a negative result is comparable for the two procedures (Figure 2).

This study has the following limits. The number of cases is relatively low, but the cases were selected with precise criteria, in order to exclude a possible bias. In particular, we excluded all the tumors that crossed the fascia, both partially and extensively. In fact, the ability to cross the deep fascia is usually related with an aggressive malignant behavior [3,23,24]. Also, all epithelial tumors were excluded.

The type of ST masses studied, in particular the fact of finding 66.7% malignant tumors, could be influenced by the centralization of suspected cases into our referral center for bone and ST tumors. For the same reason, the histology of benign lesions was relatively different from that usually observed in an unspecialized center and reported in many studies: The “simple” and more frequent subcutaneous lesions, such as cyst, lipoma, hemangioma, abscess, etc., did not arrive at our observation. For this reason, the comparison of our results with the main part of published studies has to be considered with caution.

Our choice to add the uncertain lesions at MRI, according to the literature [16], with the benign lesions, could be discussed, but we privileged the capability of MRI to identify malignant cases; instead, for CEUS, all 63 cases with histological diagnosis were assessed according to their original classification.

Furthermore, as in gray scale US, CEUS approach is operator-dependent with regard to the qualitative classification of perfusion pattern, as it is the subjective choice of ROI (region of interest) in the quantitative analysis. In fact, the result of risk indicators from a quantitative analysis of CEUS (carried out on 23 subjects) did not give significant results in differentiating malignant from benign tumors in a previous study [27]. Other authors, after the quantitative perfusion analysis, used the image representing the maximum enhancement for final assessment [26].

## 5. Conclusions

US and power Doppler are the first-level imaging techniques in the diagnostic process of ST masses. Our results introduce CEUS, combined with US, as a useful, first-level technique to discriminate benign from malignant tumors in the pure subcutaneous tissue. (We excluded the superficial masses crossing the deep fascia, as potentially aggressive malignant tumors, and pure epithelial lesions). In comparison with MRI, its sensitivity is higher, which is of fundamental importance in tumors’ first approach to diagnosis. This quality is backed up by comparable results with MRI in PPV and NPV.

The first choice of CEUS as entry tool for diagnosis in these cases could bring about a marked reduction in costs, which is a crucial problem for the management of the National Health Service. The size seems not to be a predictive factor for malignancy, because also small masses (<3.5 cm and <5 cm) can be malignant, in contrast with the international guidelines on ST sarcomas, that identify only a mass measuring >5 cm and deep in relation to the muscular fascia as potentially malignant. In fact, also the pure subcutaneous location of the mass, not crossing the deep fascia, can be potentially malignant. Further studies are needed in order to confirm our data in a larger sample and to assess the influence of any technical problem. New imaging tools and techniques will probably make easier in the future the diagnosis of malignancy in case of subcutaneous masses [35,36].

## Figures and Tables

**Figure 1 ijerph-17-08868-f001:**
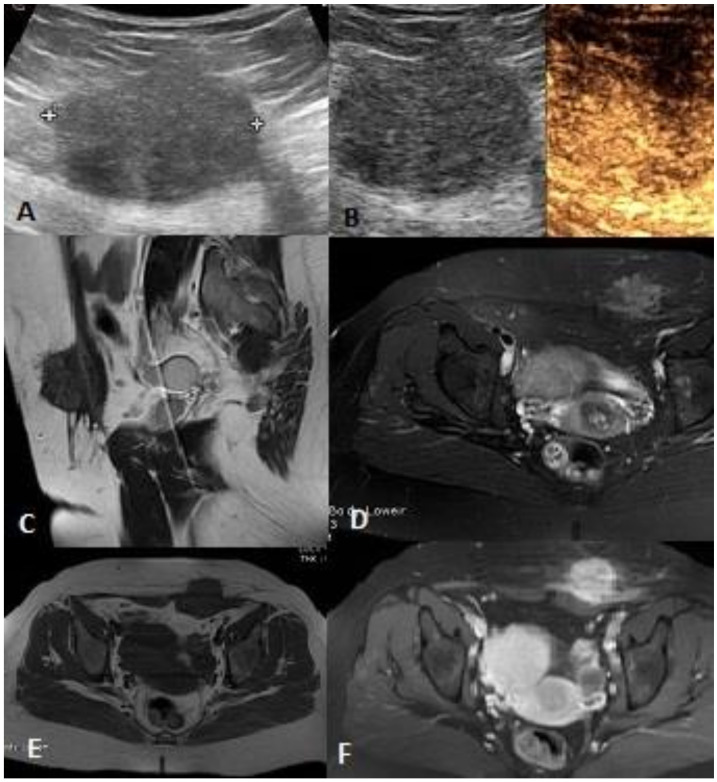
Female, 46 years old, abdominal wall endometriosis: (**A**) US demonstrates a round, heterogeneous, hypoechoic lesion with irregular edges in the deep portion of subcutaneous tissue. (**B**) CEUS reveals numerous vessels with inhomogeneous distribution and avascular areas in the center of the lesion. MRI shows a subcutaneous mass with irregular edges, heterogeneous hypointense in T1- (**C**) and T2- (**E**) weighted images, dishomogeneous high signal in STIR images (**D**) with inhomogeneous enhancement after contrast medium (**F**).

**Figure 2 ijerph-17-08868-f002:**
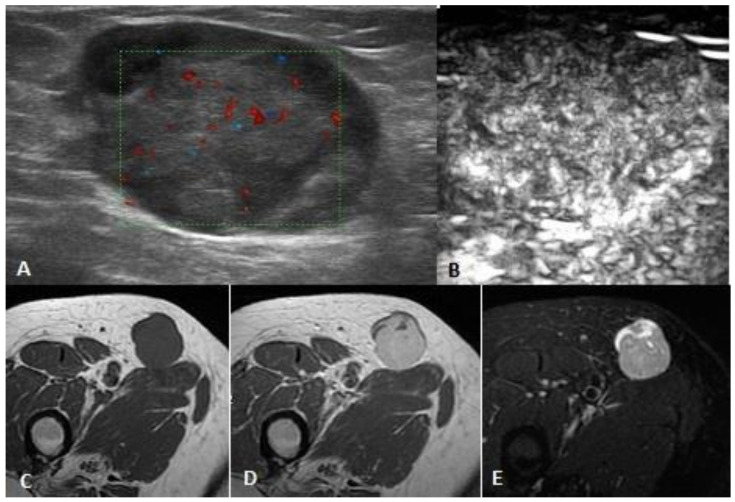
Female, 69 years old, right thigh low-grade leiomyosarcoma: US shows a round, inhomogeneous, hypoechoic, subcutaneous lesion with regular and with only few vessels at power Doppler (**A**). CEUS (**B**) demonstrates homogeneous and diffuse vascularization. MRI shows a subcutaneous mass with regular edges, homogeneous hypo-intense in T1-weighted images (**C**) and hyper-intense in STIR images (**E**) with inhomogeneous enhancement after contrast medium (**D**).

**Table 1 ijerph-17-08868-t001:** Histopathological diagnoses of the 63 soft tissue superficial tumors included in this study.

Soft Tissue Sarcoma (STS) and Non STS Malignant Tumors	Total 36
Liposarcoma	6
Primary Pleomorphic Sarcoma	6
Leiomyosarcoma	4
Dermatofibrous Sarcoma	4
Epitheliod Sarcoma	2
Myxofibrous Sarcoma	2
Carcinoma Metastasis	1
Lymphoma	5
Other malignant tumors	6
**Benign and aggressive tumors and other Benign Non-Tumoral Lesions**	**Total 27**
Lipoma and Angiolipoma	5
Schwannoma	2
Fibroma	1
Leiomyoma	1
Hemangioma	1
Desmoid Tumor, also referred to as Extra-abdominal Aggressive Fibromatosis-Intermediate tumor (benign histology, aggressive in vivo behaviour)	5
Other benign tumors	2
Cysts	1
Flogistic lesions	3
Scar tissue, post surgery reaction	3
Haematoma	1
Endometriosis	2

**Table 2 ijerph-17-08868-t002:** Univariate analysis in relation to histological diagnosis of the patients’ data and of ultrasound and power Doppler sonography characteristics.

Patient, US and PD Characteristics	All Patients (n = 63)	Benign (n = 27)	Malignant (n = 36)	*p*-Value (Ben vs Mal)
Gender (Female/Male)	30/33	13/14	17/19	0.942
Age (years)	57.9 ± 15.9	51.7 ± 15.1	62.5 ± 15.1	0.006
Location (upper vs lower part of the body)	23/40	13/14	10/26	0.097
Primary tumor/recurrent	54/9	23/4	31/5	0.917
**US and PD variables**				
Size (≤3.5/3.6–5/5.1–10/ > 10 cm)	15/17/23/8	8/8/9/2	7/9/14/6	0.579
Size (≤3.5 vs >3.5 cm)	15/48	8/19	7/29	0.348
Shape (fusiform/round/lobulated/soap_bubble/undefined)	42/6/10/2/3	20/3/4/0/0	22/3/6/2/3	0.367
Margins (regular/irregular infiltrating)	44/19	18/9	26/10	0.634
Echostructure (isoechoic/anechoic/hyperechoic/heterogeneous or hypoechoic)	2/7/6/48	2/5/2/18	0/2/4/30	0.123
Calcification (no/yes)	55/8	23/4	32/4	0.662
Vascularization (no/yes)	25/38	12/15	13/23	0.503
Septation (no/yes)	51/12	23/4	28/8	0.459
Tissue characteristcs (fat lesion/cyst/vascular lesions) ^	11/2/0	7/2/0	4/0/0	0.462
Location in relation to the fascia (protective adipofascial system, PAFS/lubrificant adipofascial system, LAFS/both)	15/14/34	6/8/13	9/6/21	0.469
Relationship with the deep fascia (no contact, contact with acute angle, contact with larger acute angle, contact with obtuse angle)	13/15/32/3	6/3/18/0	7/12/14/3	0.048

^ In 50 cases (18 benign and 32 malignant) ultrasound (US) tissue characterization was not applicable and the cases were not considered.

**Table 3 ijerph-17-08868-t003:** Univariate analysis of Contrast-enhanced ultrasound (CEUS) and magnetic resonance imaging (MRI) data in relation to histological diagnosis.

CEUS and MRI Characteristics	All patients (n = 63)	Benign (n = 27)	Malignant (n = 36)	*p*-Value (Ben vs Mal)
**CEUS variables**				
Pattern (1/2/3/4/5/6/7)	6/3/1/7/2/23/21	3/2/1/4/0/6/11	3/1/0/3/2/17/10	0.264
Pattern dichotomy (1–5 probably benign vs. 6–7 probably malignant)	19/44	10/17	9/27	0.303
Vascularisation time (no or >20′ vs. <20′)	20/43	9/18	11/25	0.815
**MRI variables**				
Signal T1 (isointense/hypointense/hyperintense/inhomogeneous)	3/49/5/4	2/20/3/0	1/29/2/4	0.227
Signal T2 (isointense/hypointense/hyperintense/inhomogeneous)	3/18/28/11	2/8/13/2	1/10/15/9	0.310
STIR (homogeneous hypointense /inhomogeneous hypointense /homogeneous hyperintense/inhomogeneous hyperintense)	7/6/27/18	2/3/10/10	5/3/17/8	0.520
Gadolinium (no enhancement/homogeneous enhancement/inhomogeneous enhancement) *	5/19/17	3/9/3	2/10/14	0.087
Peritumoral edema (no/yes)	42/20	19/8	23/12	0.697
Septation (no/yes)	46/16	23/4	23/12	0.082
Fat rim (no/yes)	18/44	5/22	13/22	0.109
Margins (regular/irregular infiltrating)	44/18	18/9	26/9	0.512

Notes: * Intravenous contrast agent was used in 41 patients

**Table 4 ijerph-17-08868-t004:** Accuracy of MRI in comparison to histology, taken as gold standard (CI 95%).

MRI	Malignant	Benign	Total
Malignant	19	7	26
Benign	17	20	37
Total	36	27	63
Sensitivity		52.8%	(37.0, 68.0)
Specificity		74.1%	(55.3, 86.8)
Positive Predictive Value		73.1%	(53.9, 86.3)
Negative Predictive Value		54.1%	(38.4, 69.0)

**Table 5 ijerph-17-08868-t005:** Accuracy of CEUS in comparison to histology, taken as gold standard (CI 95%).

CEUS	Malignant	Benign	Total
Malignant	27	17	44
Benign	9	10	19
Total	36	27	63
Sensitivity		75.0%	(58.9, 86.3)
Specificity		37.0%	(21.5, 55.8)
Positive Predictive Value		61.4%	(46.6, 74.3)
Negative Predictive Value		52.6%	(31.7, 72.7)

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
