# Peer review of "Small Subcutaneous Soft Tissue Tumors (<5 cm) Can Be Sarcomas and Contrast-Enhanced Ultrasound (CEUS) Is Useful to Identify Potentially Malignant Masses"

_ijerph, 2020, doi:10.3390/ijerph17238868_

Round 1
Reviewer 1 Report
The authors should be commended for their work.
Some minor grammatical changes are required. These were most marked in the discussion. Examples include (but not limited to): lines, 292-294, 301, 315 to name a few.
The authors appropriately highlight the limitations of small sample size. While fascinating I agree with the authors that their findings will need to be validated in a larger study.
Their rate of malignancy was higher than I expect for this population. Though they attempted to limit the bias, (excluding those that cross the fascia for example), they acknowledge that their status as a referral center naturally raises the likelihood of selection bias.
Suggest they include recommendations for role of serial imaging and timing of these in the discussion.
Author Response
Comments and Suggestions for Authors
The authors should be commended for their work. We thank the reviewer for his/her comments.
Some minor grammatical changes are required. These were most marked in the discussion. Examples include (but not limited to): lines, 292-294, 301, 315 to name a few.
Response 1. The text has been revised and grammatical changes done throughout the manuscript.
The authors appropriately highlight the limitations of small sample size. While fascinating I agree with the authors that their findings will need to be validated in a larger study.
Their rate of malignancy was higher than I expect for this population. Though they attempted to limit the bias, (excluding those that cross the fascia for example), they acknowledge that their status as a referral center naturally raises the likelihood of selection bias.
Response 2. The type of ST masses studied, in particular the fact of finding a high rate of malignant tumours, could be influenced by the centralization of suspected cases into our Referral Centre for bone and ST tumours. For the same reason, the histology of benign lesions was relatively different from that usually observed in an unspecialized center and reported in many studies; in fact, the “simple” and more frequent subcutaneous lesions, such as cysts, lipoma, hemangioma, abscess etc, do not arrive at our observation.
Suggest they include recommendations for role of serial imaging and timing of these in the discussion.
Response 3. We discussed in the first paragraphs of the Discussion the role, the advantages and limits of the different imaging techniques, starting from those acknowledged as first level investigations. We now reinforced and summarized these concepts in the Conclusions.
Reviewer 2 Report
The paper aims at assessing the accuracy of CEUS and MRI for the identification of malignant subcutaneous soft tissue tumors.
Even if the authors argue that CEUS is adequate to highlight ST tumors, it is not clear at all, from the results described in the paper, how the authors came to this conclusion.
The performances of CEUS and MRI reported in the paper are not outstanding, but it seems they are even worse than those reported in the literature (as described by the authors in the introduction). 53% of sensitivity for MRI and 37% of specificity for CEUS cannot lead to the conclusion that CEUS is "adequate". At least a combination of MRI and CEUS can have some acceptable results, but even in this case, the performances are not comparable with the reference literature.
Furthermore, it is not clear at all how the accuracy of the approaches was evaluated. In fact the authors show in table 3 that none of the image parameters they considered were significantly strong to differentiate Benign and Malignant lesions. How did the authors performed classification?
A paragraph focused on the image analysis is completely missing in the paper even though it is a crucial point. This aspect should be better detailed. A quantitative approach, rather than a qualitative one, must be implemented (e.g. radiomics indexes). Visual inspection is not sufficient to provide reliable results.
Considering all my concerns, I think the paper cannot be considered for publication in its current form.
Reviewer 3 Report
The authors proposed to distinguish a potentially malignant mass from the numerous benign soft tissue (ST) lesions. This paper can be published after addressing the following comments:
1) Can the authors detect tumor lesion smaller than 5 cm, say 2 cm?
2) What are the sensitivity, contrast, resolution, depth differences of MRI, US and PET for detecting small tumor? Because they all were used for this purpose. The characteristics will be significant for readers.
3) The authors are suggested to cite similar work: J. Nucl. Med. 61(7): 1079, (2020); Radiology: Imaging Cancer 2 (3), (2020). doi: 10.1148/rycan.2020190030
4) The authors should make the font, size, line consistent thoroughly in the whole manuscript.
Reviewer 4 Report
Reliable criteria for assessing subcutaneous lesions in terms of their potential malignancy are key in the use of US and MRI in their first approach to diagnosis. Therefore, the data obtained in De Marchi et al. study is valuable, although, some conclusions seem to be controversial.
- According to criteria provided by Grimer et al. from 2010, to which the authors have referred, the size of the subcutaneous mass less than 5 cm doesn’t exclude its malignant character. Their recommendation is “ Any patient with a soft tissue mass that is increasing in size, has a size >5 cm or is deep to the deep fascia, whether or not it is painful, should be referred to a diagnostic center with a suspected STS”. However, the 44,4% of malignant tumors measured less than 5 cm in this study, is significant enough to re-evaluate the cut-off value of 5 cm for most suspicious lesions in the future guidelines.
- The 75% sensitivity and 37% specificity for CEUS obtained in the study on 63 patients are highly insufficient to formulate the title conclusion “… CEUS is adequate to identify potentially malignant masses”.
- The authors have only provided accuracy of MRI and CEUS in comparison to histology. Similarly, it should also be counted for conventional greyscale US and power Doppler US. Is the CEUS really superior over conventional US in the screening of these lesions?
minor issues
- Were the US and MRI scans evaluated by more than one radiologist qualified in STS? Has the risk of images misinterpretation been reduced in any other way?
- Table 1. The row “Benign and aggressive tumours and other…” should be bolded and underlined
- Table 3. The last row “Hypothesis” should be explained in more detail and discussed.
Finally, this manuscript could be considered for publishing once the conclusions are re-formulated and the discussion improved.
Author Response
Please see the attachment.
Comments and Suggestions for Authors
Reliable criteria for assessing subcutaneous lesions in terms of their potential malignancy are key in the use of US and MRI in their first approach to diagnosis. Therefore, the data obtained in De Marchi et al. study is valuable, although, some conclusions seem to be controversial.
- According to criteria provided by Grimer et al. from 2010, to which the authors have referred, the size of the subcutaneous mass less than 5 cm doesn’t exclude its malignant character. Their recommendation is “ Any patient with a soft tissue mass that is increasing in size, has a size >5 cm or is deep to the deep fascia, whether or not it is painful, should be referred to a diagnostic center with a suspected STS”. However, the 44,4% of malignant tumors measured less than 5 cm in this study, is significant enough to re-evaluate the cut-off value of 5 cm for most suspicious lesions in the future guidelines.
Response 1. We agree with observation of the reviewer. Nevertheless, about 50% of the masses in our sample was smaller than 5 cm (32/63). 16 out of 36 tumours identified as malignant at histology were smaller than 5 cm, and 7 of them were smaller than 3.5 cm. As international soft tissue sarcoma guidelines identify as potentially malignant a mass larger than 5 cm and deep with respect to the muscular fascia, we intended to suggest caution in relation to the size referral criterion and a possible revision of the guidelines in the future. We modified the text (row 36 and row 278).
- The 75% sensitivity and 37% specificity for CEUS obtained in the study on 63 patients are highly insufficient to formulate the title conclusion “… CEUS is adequate to identify potentially malignant masses”.
Response 2. We followed the suggestion and replaced “adequate” with “useful”.
- The authors have only provided accuracy of MRI and CEUS in comparison to histology. Similarly, it should also be counted for conventional greyscale US and power Doppler US. Is the CEUS really superior over conventional US in the screening of these lesions?
Response 3. In the Introduction we highlighted that US and Power Doppler are not a good choice for the lesions we are working on. In fact, if they have a high accuracy in diagnosing frequent lesions like lipomas, cysts, hemangiomas, other, more “complex” masses can be misdiagnosed.
The superiority of CEUS lies in the fact that it is able to drive needle biopsy based on the vascular map (De Marchi A, Brach del Prever EM et al. Accuracy of core-needle biopsy after contrast-enhanced ultrasound in soft-tissue tumours. (Eur Radiol. 2010 Nov;20(11):2740-8.).
MRI showed a low sensitivity (52.8%) in comparison to CEUS (75%), losing about 50% of malignant tumors, that turned out to be a false negative at MRI, whereas only 25% were false negative at CEUS.
Considering that high frequency ultrasound examination is essential to detect the lesion, we believe that this first approach, performed with CEUS in a referral center, is crucial for an early detection of malignant tumours and for further patient work-up and treatment.
minor issues
- Were the US and MRI scans evaluated by more than one radiologist qualified in STS? Has the risk of images misinterpretation been reduced in any other way?
Response 1. Yes, we specified this point in the manuscript (row 158-160): “
For both CEUS and MRI the hypothesis of benign or malignant tumour or mass of uncertain origin was assessed, in accordance with the literature [2,4,11,13,19,20,22,29,30], by three radiologists with at least 10 years of relevant experience in musculoskeletal radiological imaging .”
- Table 1. The row “Benign and aggressive tumours and other…” should be bolded and underlined
Response 2. We revised the entire manuscript to uniform tables and font.
- Table 3. The last row “Hypothesis” should be explained in more detail and discussed.
Response 3. We deleted this row, as the diagnosis of benign/malignant or undefined mass at MRI compared to the gold standard (benign/malign at Histological diagnosis) is already shown in Table 4. The same (or almost the same) results reported in Table 3 can be confusing for the readers.
Finally, this manuscript could be considered for publishing once the conclusions are re-formulated and the discussion improved.

Round 2
Reviewer 2 Report
The authors tried to address the concerns I raised in my previous review, however, the paper was not substantially improved and a robust quantitative approach in image analysis is still missing.
Reviewer 4 Report
The authors' responses as well as the present form of the manuscript are satisfactory. I only have one suggestion: the conclusion (abstract line 33) - „CEUS may help in the differentiation of benign and malignant purely subcutaneous soft tissue tumours” is unjustified and should therefore be removed. Only 37/63 of the cases were correctly classified by CEUS as benign or malignant. Furthermore, none of CEUS characteristics has come out by itself as indicator of benignity or malignancy.
Author Response
„CEUS may help in the differentiation of benign and malignant purely subcutaneous soft tissue tumours” is unjustified and should therefore be removed."
We removed this sentence, as suggested by the reviewer.
We thank the reviewer for this suggestion and for those of the first round.